# Medical Staff Shortages and the Performance of Outpatient Clinics in Poland during the COVID-19 Pandemic

**DOI:** 10.3390/ijerph192214827

**Published:** 2022-11-11

**Authors:** Piotr Korneta, Magda Chmiel

**Affiliations:** 1Faculty of Management, Warsaw University of Technology, 02-524 Warszawa, Poland; 2Faculty of Chemical Engineering and Commodity Science, Kazimierz Pułaski University of Technology and Humanities, 26-610 Radom, Poland

**Keywords:** SARS-CoV-19, pandemic, multiple case study, outpatient clinics, medical staff shortages, public aid, dentistry, primary healthcare

## Abstract

Unlike many industries, healthcare was simultaneously affected by the COVID-19 pandemic in two opposite ways. On the one hand, the industry faced shortages and overload of many medical representatives such as nurses and infectious disease professionals, but on the other, many medical professionals such as dentists were left with considerably reduced demand. The objective of this paper is to study the efficiency of medical staff allocation and the performance of small and medium sized outpatient clinics in Poland during the COVID-19 pandemic. Owing to the contemporaneity of this problem, we have employed a multiple case study approach. Our sample comprises 5 small and medium-sized outpatient clinics located in Poland in the 3-year period 2019–2021. The results indicate a considerable percentage of medical staff employed in small and medium-sized outpatient clinics remained outside the pandemic, despite their potential provision of healthcare services. Four of the five clinics studied remained passive towards the pandemic. In view of future pandemics, the indications we provide have practical implications for outpatient clinics executives and public health policymakers.

## 1. Introduction

The unfolding COVID-19 pandemic has resulted in a significant health and economic crisis around the world [1,2]. Such an exogenous shock has considerably altered the competitive landscape for the majority of companies [3,4]. Businesses entities around the world have reported decline in sales [2,5], difficulties in operations [6], and increased difficulties in accessing capital [7]. Scholars have also already noted an increased number of companies bankrupting [8]. In order to decrease the negative impact of the COVID-19 pandemic on businesses, governments in many countries have implemented various means of public subsidy [9,10,11]. Despite a considerable deterioration in the business environment for most companies, some companies managed to improve their performance during the pandemic. Scholars have already explained how COVID-19 could become an opportunity to successfully transform companies [12]. Other researchers have postulated the implementation of a mix of cutbacks and strategic investments with significant cost discipline, and development of opportunities offering reliable returns during the pandemic should enable companies to improve their performance [13,14]. It is also acknowledged that companies deeply rooted in advanced technologies are more resistant to the COVID-19 pandemic [15]. Similar to other industries, the COVID-19 pandemic has been a very challenging period for the whole healthcare industry. Many hospitals have undergone considerable adjustments implementing infection prevention and control measures (IPC). Although the scope of IPC differed greatly by country and by sector, it included procurement of commodities, including personal protective equipment and hygiene and cleaning supplies, governance and infrastructure improvements, interventions for COVID-19 emergency response, implementation of portable high-efficiency particulate absorbing (HEPA) filtration devices, installation of exhaust fans and provision of temporary hand hygiene stations in isolation wards. Sometimes longer-term, sustainable improvements such as natural or mechanical ventilation and sustainable water, and sanitation and hygiene services in health facilities were included [16]. In addition to IPC implementation, hospitals have had to set up more intensive care units and expand their capacity. Primary healthcare service providers had to protect patients and themselves from potential infection. This required a change in their practice patterns in line with new hygiene requirements, making greater use of personal protective equipment, provision of increased volumes of disinfectants, and enabling patients to maintain distance in waiting rooms. Furthermore, many healthcare entities frequently had to take on new tasks, such as COVID-19 testing [9,17]. Many scholars also noted the accelerated development of telemedical services during the pandemic, seen not only in large healthcare businesses, but also in small clinics [18,19,20].

We study the healthcare industry because of two major reasons. Firstly, because this sector plays a critical role in responding to the pandemic. Secondly, because the healthcare industry was simultaneously affected by the pandemic in two contradictory ways, i.e., the huge new demand for COVID-19 treatment and diagnostics, overloading the infectious disease, internal medicine, and acute care wards of hospitals, while the demand for other medical services decreased considerably (e.g., dentistry or dermatology), as patients delayed their medical treatment. Hence, we initially assumed some segments of the healthcare industry experienced the same fall in revenues as many other non-healthcare businesses, while other segments of the industry experienced unexpected rises in revenues. Managing increased demand for infectious disease treatment requires more premises, equipment, and medical staff. Proper management of the latter appears to be the most challenging. Moving medical staff between different healthcare segments, although possible, is not a straightforward task. Given this, the pandemic provides a unique opportunity to study how clinics where services declined during the pandemic reacted to the crisis. These clinics had 2 options, either to shift into new activities, learning new skills, and accepting higher levels of infection risks or avoid the pandemic and accept a temporary decline in revenues and as a consequence economic loss. The healthcare industry had already reported significant medical staff shortages before the pandemic [21,22,23], with researchers postulating a further increase in these shortages during the pandemic [24]. In this paper, in line with majority of scholars, we define the healthcare staff shortage as a situation in which healthcare providers are unable to hire medical personnel at market salaries as to achieve desired staffing levels [25]. This paper adds to the literature, covering the initial understanding of the reasons behind medical and managerial decisions, alongside their financial consequences for those outpatient clinics which were not on the frontline against the pandemic.

The objective of this paper is therefore to study the efficiency of medical staff allocation and the performance of small and medium-sized outpatient clinics in Poland during the COVID-19 pandemic. In order to achieve this, we employ a multiple case study, which comprises five outpatient clinics in Poland. Our case study approach consists of 19 in-depth interviews conducted in two rounds, a review of administrative documents, and the analysis of company financial results in the period from 1 January 2019 to 31 December 2021. For several reasons, the scope of this paper has been deliberately narrowed to small and medium-sized enterprises (SME). Firstly, SMEs are more prone to being severely affected by economic disruptions [26]. Secondly, SMEs have lower capabilities and resources than large companies, so from this perspective they require more support from researchers [27]. Thirdly, SMEs are more flexible when threats or opportunities appear in their environment than large bureaucratic organizations [28], and finally, because SMEs account for approximately 95% of world organizations, hence their role is critical in each country’s economy [29].

## 2. Materials and Methods

To study how the COVID-19 pandemic affected small and medium-sized outpatient clinics in Poland together with medical staff allocations, we employed a multiple case study approach. The case study is commonly acknowledged to be best suited to studying complex, real-life phenomena for which theoretical knowledge does not exist or is scarce [30]. We chose to include multiple cases to compare clinics which engaged significantly in COVID-19 activities with those which did not engage in any pandemic-related activity, so that we could understand the pandemic’s impact on the two extremes of outpatient clinics. Also, the use of multiple case studies instead of a single case study increases the robustness of the findings [31,32].

### 2.1. Sample

We used purposive sampling [33] in this study and selected companies which approached the COVID-19 pandemic in two completely different ways. The first group comprises a medium-sized outpatient clinic which engaged significantly in COVID-19 healthcare activities, while the second includes four small outpatient clinics which undertook no COVID-19 healthcare activities and thus provided the same services before and during the pandemic. Since most outpatient clinics provided no COVID-19-related activities, the first group consists of only one company, while the latter is more numerous. In selecting the companies, the focus was on the following criteria: (1) considerable experience in the healthcare sector; (2) small or medium size; (3) age of company over 10 years; and (4) number of employees above 10, of which at least 5 were medical doctors. As already indicated, this study is focused on SME; however, we did not take into consideration self-employed businesses and other companies with fewer than 10 employees, as it is believed that management of such entities is not complex enough to be studied. Organizations below 10 years old were not studied in order to avoid the influence of difficulties and challenges characteristic of newly established companies being formed during this period.

### 2.2. Cases Description

In Appendix A, a summary of the most important information relating to each of the 5 cases selected for the study is provided.

#### 2.2.1. Case 1

The first case is a clinic which was significantly engaged in several COVID-19 healthcare activities (testing, vaccination, and dentistry for infected patients) during the pandemic. This is a medium-sized clinic located in two major districts of Poland. It provides healthcare services in 12 locations: 9 in Warsaw, 2 in Radom and 1 in Kielce. Its core business, prior to the pandemic, comprised mostly dentistry, primary, and specialized healthcare. In the 5 years prior to the study period, its sales revenue grew around 20% per year. The company is run by the only shareholder and 2 managers. Immediately prior to the COVID-19 pandemic, Case 1 introduced mobile dentistry to its portfolio, aiming to treat children at primary schools.

#### 2.2.2. Case 2

The second case, in terms of scope of sales, is very similar to Case 1, consisting of primary healthcare, dentistry, and specialized health care. This second company, however, is considerably smaller than the first and undertook no COVID-19 activities during the pandemic. The clinic employs 3 general practitioners, 1 pediatrician, 10 dentists, 3 nurses, 5 dental hygienists, and 3 back-office employees. The clinic is located in Warsaw and is run by one manager with very limited support from one of two shareholders.

#### 2.2.3. Case 3

The third case is focused solely on dentistry. Its core specialty is prosthetics, followed by orthodontics and dental surgery. Since general dentistry has the smallest share of the company’s sales, and due to the high-class specialists providing services there, the clinic is perceived by patients as being of high quality. This company is owned and run by two medical doctors: a dental surgeon and a dentistry prosthodontist. This clinic undertook no COVID-19 activities during the pandemic. Case 3 is located in Nowy Dwór Mazowiecki.

#### 2.2.4. Case 4

The fourth case is a dental surgery located in Gdańsk. This is a premium clinic specializing in implants. It is managed by the only shareholder and his daughter. The only shareholder is also a key dental surgeon at that company. The clinic employees 7 medical doctors. Prior to the pandemic, the company introduced new, more advanced implants to its portfolio. This clinic also undertook no COVID-19 activities during the pandemic.

#### 2.2.5. Case 5

The fifth case is a dental clinic specializing in general dentistry. Around 50% of the company’s sales are financed through the National Health Fund so that 50%, i.e., only 1 out of 2 patients pays out of pocket. The company provides general dentistry services to adults and children, with very limited specialized dentistry services. The clinic is run by a young manager with only a few years of experience in the healthcare industry. The clinic employees 8 dentists, 7 dental hygienists, and 3 back-office employees. The shareholder of the company pays limited attention to the company as they are primarily focused on their other businesses. Case 5 has not undertaken any COVID-19 activities during the pandemic.

### 2.3. Data Collection

#### 2.3.1. In-Depth Interviews

Interviews are frequently used in studies and have been widely acknowledged as one of the most important methods of collecting qualitative data [34,35]. Given the novelty and complexity of the area studied, we consider in-depth interviews to be the most suitable choice for this research. In particular, due to considerable shortages in the literature on COVID-19′s impact on outpatient clinics, it was not possible to formulate the right set of questions for structured interviews. Hence, in-depth interviews were employed. Such interviews are based on comprehensive discussions between the interviewer and interviewee. They have an overriding purpose prompted by the research conducted and are thus strongly guided by the interviewee’s perceptions and experiences [36]. Before the in-depth interviews commenced, the interviewer was allowed adequate preparation, which, inter alia, included familiarizing himself with the publicly available information concerning each of the 5 cases studied. The right preparation for in-depth interviews is critical for successful investigation, as it underpins the interview process and influences subsequent stages of the research [37].

The interviews were held in the subject’s premises, or in the accounting office. Prior to the in-depth interviews, voluntary consent from each of the interviewees was obtained. The interviews were conducted in two rounds. In the first round, 14 interviews were conducted in total. These interviews were held between June and October 2021. The first 6 interviews were conducted with case 1. Since this case differed significantly form the remaining 4, the number of interviews were increased. Hence, in case 1 we have interviewed: (1) the owner and the operations manager, (2) the strategy and business development manager, (3) the chief accountant, (4) the recruiter and (5–6) two local managers responsible for operations. Next, we conducted 8 in-depth interviews with the representatives of the remaining 4 cases. In each of cases 2–5, the person running the company, either shareholder or manager, was interviewed. In the second round of interviews, held in April and May 2022, 5 follow-up interviews, one for each case, with the persons already interviewed during round one, were conducted.

#### 2.3.2. Financial and Non-Financial Documents

The study uses a significant quantity of financial data. During the two rounds of interviews, the representatives of each case were asked to obtain trial balances for the 3-year period of the study, beginning 1 January 2019 and ending 31 December 2021, from accounting departments. The financial data were in Polish zlotys (PLN). We translated PLN into euros (EUR) using the European Central Bank average exchange rates for 2020, which amounted to 4.4423. The financial data for each of three years studied were converted with the same exchange rate. Although the exchange rates varied over the study period, only one rate was employed in order to avoid the impact of exchange rate differences on our study. For indicative purposes only, the European Central Bank average exchange rates for PLN into EUR in 2019 and in 2021 were 4.109 and 4.5652, respectively. Additionally, inflation rates (year on year) are provided for each year of the study.

Finally, during the two rounds of interviews, basic administrative documents were provided for each of the 5 studies relating to their internal regulations and procedures, as well as several healthcare industry presentations.

### 2.4. Data Analysis

The results were obtained from the following three data sources for each of the cases studied: (1) in-depth interviews, (2) financial documents, and (3) other documents. The use of mixed methods in qualitative studies improves the robustness of the results and has been used previously in other studies [38,39]. Horizontal analysis was employed to study the changes in financial results between the two pandemic years (2020 and 2021) and the year before the pandemic (2019). Such analysis allows the relative changes in the different variables to be studied over time. The horizontal analysis for each of the 5 cases investigated the following 3 groups of financial measures: (1) revenue, including sales from dentistry, primary healthcare, specialists (otolaryngologists, cardiologists and others) and other revenues specific to the case or the pandemic, such as COVID-19 testing, dentistry; (2) medical costs, including the salaries of medical doctors or medical personnel (nurses, dentistry hygienists) and medical materials, and (3) other running costs (rental, maintenance, administrative and miscellaneous). The difference between the financial data for 2021 and 2019 was calculated and the percentage change over time was quantified.

The results obtained from the interviews were used to corroborate the patterns that evolved from the financial and administrative data collected, so that the validity of the findings could be enhanced. A similar approach has already been employed in other studies [40].

## 3. Results

### 3.1. Case 1

Table 1 shows the summary of Case 1 financial performance for the 3 consecutive financial years ending 31 December 2019 to 31 December 2021; an absolute and a percentage change in the selected variables within the 3-year period, profit before tax presented as a percentage of sales and inflation rates for each of studied years.

The first company decided to explore all of the COVID-19 business opportunities for the healthcare industry during the pandemic period. All 6 interviewees indicated that the scope of the clinic’s work has considerably shifted from its core activities, including mostly dentistry and primary healthcare, into COVID-19-related activities. At the beginning of the pandemic, Case 1 obtained a minor value subsidy from public aid and later a following one from the “Anti-Crisis Shield” of around EUR 82 thousand in the form of a loan to be partially repaid. The volume of such government support was too low to prevent losses for the company. The strategy and business development manager stated: “*At the very beginning of the pandemic, we very quickly realized that either we evolve into COVID-19-related healthcare activities or we would have to dismiss a considerable number of our employees. This was mostly because we faced a sudden drop in patients visiting our clinics and the reluctance of medical staff to provide any services within our premises”*. He also added that in the first 3 months of the pandemic the National Health Fund and local authorities asked outpatient clinics to launch anti-COVID-19 activities (testing, dentistry to patients in quarantine), with very few or no clinics, depending on region, willing to cooperate with the government against the pandemic. He concluded: “*The fear of personnel infection was so high that to gain new government contracts for provision of services had never been so easy. For the first time, we had almost no competition in obtaining new healthcare contracts*”. The decision of the Case 1 management was to enter every COVID-19 healthcare activity the clinic had the competencies to serve, or which could develop within a reasonable period, such as COVID-19 PCR testing. The representatives of Case 1 stated the company had no major problems switching activities. In the case of COVID-19 testing, the laboratory Case 1 supplied PCR tests to came with substantial help (training of personnel, provision of information, and so on). And so, Case 1 moved nurses, midwives, and dentistry hygienists to PCR testing. After the implementation of new practice patterns resulting from significantly higher hygiene requirements and ensuring the right protective equipment, selected dentists accepted the provision of dentistry services to potentially infected and quarantined patients. The owner of the company and the recruiters stated that the hardest part of redirecting the company to pandemic activities was to ensure there were the medical staff to provide the services. They said there were doctors and dentists who refused to provide any services in the first months of pandemic under any circumstances. The recruiter provided an example where she spoke 3 times with one dentist who before the pandemic accepted any activity, including treatment of children in care homes with reduced fee and so on, but refused to provide any services to patients in pain in quarantine. Finally, the dentist confessed to having cancer and being on medication which prohibited her from taking any risk of COVID-19 infection. Another group of medical staff required extensive explanations and needed to be asked or even begged by the recruiters to provide services to potentially infected patients. Finally, the last group of medical staff required no special attention. They were simply willing to do their jobs, treat the patients, and appreciated the fact that potential infection is part of their profession. Not all staff accepted this strategy, however. Several doctors and one regional manager did not accept this approach and left the company. Due to the newly developed COVID-19 business segments, the company sales grew by 39% in 2020 as compared with 2019. According to the management, in 2021 the company was almost entirely focused on COVID-19 activities, which accounted for 70% of total sales revenue. Since the sales margins on COVID-19 testing and dentistry were above 50%, Case 1 recognized very high profits. The profits gained on COVID-19 activities were reinvested in the acquisition of 3 dentistry clinics in 2021 and improvements (renovations and new medical devices) in their existing clinics.

### 3.2. Case 2

Table 2 shows the summary of Case 2 financial performance for the 3 consecutive financial years ending 31 December 2019 to 31 December 2021; an absolute and a percentage change in the selected variables within the 3-year period; and profit before tax presented as a percentage of sales and inflation rates for each of studied years.

Case 2 is very similar to Case 1 in terms of its sales structure. This clinic provides no advanced services, so it has many competitors in Warsaw. This clinic approached the pandemic in a completely different way. Case 2 ceased its dentistry and specialist operations in spring of 2020, when the fear of infection reached its peak in the population. Primary healthcare provided telemedical services through the height of pandemic. Case 2 engaged in almost no pandemic-related activities, with the exception of vaccinations against COVID-19 in late 2021. The manager of the clinic stated this decision was based on two major factors. Firstly, there were serious shortages of medical doctors and dentists in Warsaw, and since the majority of medical personnel was not willing to work during the pandemic, the company did not want to force them, i.e., Case 2 was afraid of losing employees. Secondly, the company employs a mix of personnel, with a significant share of employees who are over 60 years old. If they had been infected with COVID-19, the risk of death was higher than the population average. The manager of Case 2 stated: “*the decision to protect medical personnel and to cease service provision to our patients was one of the toughest decisions I made in the whole my life*. *I knew very well that the job of the clinic is to treat infected patients accepting the risk of potential infection, but given the average age of our medical staff I decided to protect their lives instead of protecting the health and lives of our patients”*. The clinic lost a number of patients to competitors who remained open through the pandemic. Case 2 did not even attempt to ask medical staff to provide COVID-19 activities; however, the Case 2 representative stated one doctor and two nurses moved voluntarily to a COVID-19 temporary hospital as an additional assignment. We asked the Case 2 manager if any employees in the dentistry division could have considered a transfer to other clinics engaged against the COVID-19 pandemic, and we were told that: *“it’s very likely that 2 or 3 younger employees would consider such a transfer, but I honestly do not know anything about such opportunities for dentists”*. In the first months of the pandemic, Case 2 obtained a public subsidy from the Anti-Crisis Shield of around EUR 37 thousand in the form of a loan to be partially repaid. Despite this aid, sales and profitability decreased in both 2020 and 2021. The increase in remaining running costs in 2021 was due to inflation, which the company could not offset by decreasing sales.

### 3.3. Case 3

Table 3 shows the summary of Case 3 financial performance for the 2 consecutive financial years ending 31 December 2019 and 31 December 2020; an absolute and a percentage change in the selected variables within the 2-year period, profit before tax presented as a percentage of sales and inflation rates for each of studied years.

At the very beginning of 2019, Case 3 acquired new medical equipment (an intraoral scanner and a pantograph with software) under lease contracts, expecting its sales to increase. With the sudden appearance of the SARS-CoV-2 pandemic, the company limited its activity. During spring 2020, the company only provided services to selected pain patients. One of the shareholders stated: *at my age I considered the potential COVID-19 infection consequences as too serious, so I preferred to protect my life instead of the business*. Next the doctor explained that in dentistry, unlike other professions, the risk of contamination is especially high. This is because during dental treatment oral epithelia and saliva are sprayed. Both shareholders stated that during the COVID-19 pandemic there were no business opportunities for dental clinics, so they did not consider any of them. Case 3 received a public subsidy of EUR 22 thousand from Anti-Crisis Shield in the form of a loan to be partially repaid. Although this subsidy covered part of the running costs, it was not enough to turn the company profitable and, even more importantly, to cover monthly leasing payments. Since one of the two shareholders were just divorced and could not invest any extra funds in the company, they decided to sell the company and quit the business. At the end of December 2020, Case 3 was sold to a major healthcare company.

### 3.4. Case 4

Table 4 shows the summary of Case 4 financial performance for the 3 consecutive financial years ending 31 December 2019 to 31 December 2021; an absolute and a percentage change in the selected variables within the 3-year period; and profit before tax presented as a percentage of sales and inflation rates for each of studied years.

The majority of Case 4 sales are dental surgery and implant services. In 2019, the clinic introduced advanced, high margin implants. In 2020, Case 4 did not engage in any COVID-19 activities, stating: “*COVID-19 activities is not our business. Our job is to insert implants and provide the highest quality dental surgery services in Gdansk, or even nationwide”*. The company was very well aware of the deterioration in financial results and fully accepted this fact. The shareholder stated: “*I have enough funds to close the business, pay my employees’ salaries and do nothing for 2 years, so I will not consider any COVID-19 activity”.* Next, we discussed the potential loss of patients to competitors who remained open through the pandemic. They explained that the dental services they provide are highly advanced with no true competition in their region. Besides, the clinic’s brand and the names of the surgeons were well recognized and trusted by patients in the Gdansk region. They concluded that if they ceased provision of services, the patients would wait until they reopened, noting that a lack of implants was not a threat to patients’ lives. We also asked the shareholder if he considered any of his employees would like to work against the COVID-19 pandemic. He stated: “*in my clinic I do not see the need for my employees to engage in any COVID-19-related activities. They are on a payroll.* However, he added: *I know that hygienists from several other clinics in Gdańsk, especially those employed on civil contracts instead of labor agreements, engaged in PCR COVID-19 testing of potentially infected patients*”. As a result, of this approach, the sales of Case 5 decreased in 2020, as compared with 2019 by 24%, and the company recognized almost no profit. In 2020, Case 4 obtained a public subsidy of EUR 25 thousand from the Anti-Crisis Shield in the form of a loan to be partially repaid in 2021; however, in line with the expectations of the dental surgeons, sales increased sharply. In 2021, Case 5 sold a significant quantity of high-margin implants (introduced to the portfolio in 2019), which were not sold throughout 2020. The lower profitability in 2021 than in 2019, despite higher sales, was due to high inflation.

### 3.5. Case 5

Table 5 shows the summary of Case 5 financial performance for the 3 consecutive financial years ending 31 December 2019 to 31 December 2021; an absolute and a percentage change in the selected variables within the 3-year period; and profit before tax presented as a percentage of sales and inflation rates for each of studied years.

Following 3 unprofitable years and a change of manager in 2018, Case 5 recognized growth in sales and profits in 2019. During the pandemic, Case 5 was open to patients, including the spring of 2020, when the fear of infection was the highest among the Polish population. To ensure the safe provision of services, the clinic invested heavily in anti-infection materials and reorganized the flow of patients. Although by management decision the clinic was open for patients, most dentists, due to the high risk of infection, refused to work. The manager of the clinic stated that in those most risky months the clinic operated 2 afternoons instead of 5. The manager said: “*the key problem was the medical doctors. As for the hygienists however, I would state the opposite. The ladies were asking me for additional hours, which I could not give them. They needed to earn but there was no job for them, neither in our clinic, nor in other dental clinics*”. Although less than expected, in contrast to the majority of competitor clinics, Case 5 remained open for patients. The manager said she did not want to force the doctors to come to work as she was afraid that if she pushed them hard, they would quit their jobs. She said: “*several doctors openly stated they would rather quit their job than provide services to potentially infected patients*”. Thus, despite the fact she asked them politely to keep providing their services, she was not in a position to ensure continuity of treatment. The decrease in sales and the loss for the period in 2020 resulted from considerable absences of dentists. Case 5 received a public subsidy of EUR 23 thousand in 2020 from the Anti-Crisis Shield in the form of a loan to be partially repaid. According to the manager, this aided in keeping up financial liquidity. This was especially important, as the clinic incurred many considerable pandemic costs, such as increased protective materials, antiseptic costs, etc. Case 5 did not consider implementing any pandemic-related activities such as COVID-19 testing or vaccination, but through the whole study period kept focused only on to its core business, i.e., dentistry. According to the Case 5 manager, the considerable increase in sales in 2021 was due to fact that the clinic remained open to patients during each month of the pandemic, which was appreciated by the patients and also because good relationships were maintained with medical staff by the clinic.

## 4. Discussion

In general, the onset of the COVID-19 pandemic resulted in considerable declines in the economies of a majority of countries [2,5]. The revenues and profits of many companies decreased significantly [5], with some going bankrupt [8]. Several scholars reported massive lay-offs caused by the pandemic in various industries [41]. It has already been indicated by researchers that this deterioration in the business climate during the COVID-19 pandemic reflected 3 major underlying reasons. Firstly, the different forms of isolation, including lockdowns, quarantine, or shutdowns of events and corporate offices, imposed by the majority of governments, resulted in a decrease in services provided, goods produced, and consumer spending. Secondly, the uncertainty about the situation and its further development led to a significant decrease in consumer spending [2,42]. Thirdly, the change in consumer behavior and their needs decreased spending [43]. Despite the considerable decrease in results obtained by most companies, some businesses managed to explore business opportunities arising from the unique situation caused by the COVID-19 pandemic. It has been already reported that these companies have been innovative, flexible, and naturally resilient [31,44]. Others have already indicated that strong leadership competencies played a key role in rapid and successful COVID-19 business transformations [1,45]. For this study, 5 small or medium-sized outpatient clinics were selected whose core activities were not related to infectious diseases. We analyzed their performance in the 2019–2021 period comprising the 2 years of the pandemic. In the first year of COVID-19 pandemic, i.e., in 2020, the results of 4 of the 5 clinics studied deteriorated considerably. The only clinic whose results improved in 2020 as compared with 2019 was Case 1, which adjusted its business model to the COVID-19 pandemic. In the 5 sample outpatient clinics studied, 4 of them did not adjust their business models to the COVID-19 pandemic. Furthermore, these 4 clinics perpetually and significantly reduced provision of services in the first year of the pandemic. The rationale for this approach was the same as for non-healthcare businesses—including lockdowns, quarantine, and the overall fear of employees becoming infected with COVID-19. This approach led to very poor financial results for these clinics, including the liquidation of one of them. The results obtained are therefore aligned to many studies that reported mass deterioration of company financial results during the pandemic [2,5]. The only outpatient clinic (Case 1) which improved its financial results in 2020 as compared with 2019 was the clinic which significantly engaged in provision of services related to COVID-19 (testing, vaccination, and dentistry for infected patients). From the very beginning of the pandemic, Case 1 commenced provision of COVID-19 tests and dentistry. Neither of these two tasks was initially straightforward, and strenuous efforts were required from the Case 1 executives. Case 1 had to, *inter alia*, set up new safety procedures, ensure sufficient supplies of COVID-19 tests, and in general overcome the fear its employees had of providing services to potentially infected patients. Due to Case 1′s agility and strong leadership skills the company expanded its activities and improved financial results. The results for Case 1 obtained in this study endorse the postulates of other researchers [12,13,14] who claim the COVID-19 pandemic created not only threats, but also business opportunities for those agile companies which managed to transform their businesses.

Over an extended period, healthcare staff shortages have been widely discussed in the literature, with many researchers indicating this problem to be omnipresent [21,22,23]. Research has already indicated that, due to additional patient loads during the pandemic, the aforementioned shortages have increased further [24]. Since a considerable quantity of evidence on significant medical staff shortages has already been reported in this paper, an analysis was conducted of whether the medical staff of small and medium-sized outpatient clinics in Poland were properly allocated during the pandemic. In this study, we have analyzed medical staff performance during the pandemic, paying particular attention to medical staff shortages and their allocation for various pandemic tasks. As with the whole healthcare industry, each of the 5 cases studied experienced medical staff shortages before the COVID-19 pandemic. In the first year of the pandemic, however, 4 of the 5 cases studied did not engage in any pandemic activities and reduced their service provision. This reduction was because of the fear of personnel infection and the fear of patients of being infected at the clinic. It was identified that the approach of the medical staff of small and medium-sized clinics towards working during the pandemic can be divided into three different groups. The first group comprises medical staff unwilling to work during the pandemic. The representatives of this group are better off, so they can afford to cease service provision, or of weaker health, i.e., elderly or with immune system deficiencies. Their constitutional right is not to work, which must be accepted. The second group comprises those who were willing to work during the pandemic and so support the lives and health of society. This group on their own relocated to hospitals and medical points where they could work with infected or potentially infected patients. This group primarily includes nurses and general practitioners. The third group consists of medical staff willing to provide medical services during the pandemic, but due to lack of experience in infectious diseases they did not engage in any COVID-19 activities. This group includes dentists, hygienists, medical doctors with other specializations such as dermatologists, radiology technicians, midwives, physiotherapists, and others. This group therefore comprises persons with varying medical education. In Poland, dentists study for 5 years, and during their course they have classes on infectious diseases, pharmacology, and internal medicine, so initially it appears possible, following some update training, to move them to certain fields of pandemic healthcare activities. But not only them, other medical staff such as midwives or physiotherapists could also work as additional workforce in healthcare service provision during the pandemic. Thus, the third group of medical staff, who were willing to work but did not provide or provided significantly fewer services than they could, presents unused reserves of medical staff during the pandemic. The results of this study confirm other studies which claim great medical staff shortages during the COVID-19 pandemic [24]. However, this study adds to the literature that we have identified a group of medical staff employed and small and medium sized outpatient clinics not specializing in infectious diseases, who would like to work with infected or potentially infected patients but did not do so. Although the reasons for not providing the services by this group are not homogenous, it appears if the government or local authorities had provided additional training, incentives or clearly communicated initiatives, the volume of medical staff working against the future pandemic could have increased noticeably.

In order to offset the negative effects of the economic crisis caused by the pandemic and the restrictions imposed on inhabitants of many countries, such as lockdowns or quarantine, the governments supported the companies and sometimes natural persons with public aid. The objective of public aid was primarily to limit the loss of income of companies and to maintain employment levels. The range of solutions employed by public subsidy was very broad and varied according to: state, industry, company size, and over time. Among the frequently applied forms of public support were: (1) non-refundable transfers of funds paid to enterprises, (2) guarantees to enable commercial borrowing or loans, or to subsidize costs of external financing, (3) state transfers that will have to be repaid but on preferential terms, (4) various fiscal instruments such as exemption from paying public levies, accelerated tax refunds, and other tax allowances, (5) deferral of deadlines to meet public obligations [10]. In Poland, public aid was introduced under the name of the Anti-Crisis Shield, with successive versions announced regularly. This program has several sections and addresses all companies, from micro (fewer than 9 employees) to large. The companies were subject to this program if in two months their sales revenue decreased, which was a straightforward criterion to meet for the bulk of Polish companies, and so many companies benefited from this program. A significant portion of the Anti-Crisis Shield related to subsidies to be partially repaid, non-refundable loans if the company did not lay off its employees over the succeeding 3 months and loan guarantees up to PLN 3.5 million (EUR 0.8 million). It has been already postulated that the Anti-Crisis Shield on the one hand prevented many companies from going bankrupt, but on the another caused a distortion of the pre-pandemic economic system [11]. As for the healthcare industry, the governments of various countries had to focus on two major aspects of public aid. The first was how to compensate hospitals and outpatient clinics for the loss of revenues related to reduced activity. The second was how to pay for COVID-19 related services and the costs of higher hygiene standards. The latter was especially important, as healthcare providers had to: change their practice patterns in line with new hygiene requirements, make greater use of personal protective equipment, provide disinfectants, enable patients to maintain distance in waiting rooms, take on new tasks such as COVID-19 testing, and provide more home visits and remote services to protect patients and themselves from potential infection [9,17]. Public aid for the healthcare industry appears to have been especially important as it could potentially impact treatment patterns, admission policies, activity levels, efficiency, and quality of care [46,47,48]. It has been already postulated that policymakers should increase the resilience of payment systems during future pandemics. This entails: (1) having the right flexible systems in place in order to be able to adjust them rapidly; (2) being aware of the economic incentives created by these adjustments, which may cause risk selection or overprovision of care; and (3) periodically evaluating the effects of payment adjustments on access and quality of care [9]. In this paper, we have studied 5 small and medium-sized outpatient clinics in Poland not operating in the infectious disease area during the COVID-19 pandemic.

Each of the cases studied received public aid in the form of a subsidy from the Anti-Crisis Shield to be partially repaid. Case 1, despite very high profits realized during the pandemic, obtained public aid to be repaid only in part as it maintained (increased) employment level. Case 2 received a donation, which improved its financial situation; however, Case 2 could easily engage in pandemic-related activities at a time when there was a huge shortage of medical staff. This is especially important, as Case 2 employs general practitioners and nurses. The public aid received by Case 3 was not enough to prevent its liquidation. Case 4 openly stated they did not need any public aid, as they were better off and could survive for two years without providing healthcare services. Case 5 received a donation, which a was great help, but did not even consider a shift to pandemic-related healthcare services. The results obtained indicate that the public aid obtained by the 5 cases was not efficient and did not play its role correctly. Hence, the conclusions of this study are aligned with conclusions from other studies, which claim policymakers should increase the resilience of payment systems during future pandemics [9]. Furthermore, given that one of the cases significantly increased its profits during the pandemic, while another went bankrupt, our results endorse the conclusions of others who claim that public aid caused a distortion of the pre-pandemic economic system [11].

We shall note, however, that this study has several limitations, resulting primarily from the qualitative methods applied. The multiple case study approach does not allow for an empirical generalization in probabilistic or deterministic terms, whereas our results should be considered as ideas that provide reasonable expectations of similar results in other cases. The results obtained in case studies should be therefore validated or falsified by quantitative research [49,50]. Secondly, since each of the 5 cases was located in Poland, the results obtained in this study might not be applicable to other markets or other countries. However, as initially assumed, the objective of this study was to identify phenomena unique to the COVID-19 pandemic, which could not be identified using different scientific methods. As a result, the indications identified are a good pointer for further research, which could quantify the volume of medical staff which could have been engaged against the pandemic, but was not. Another indication for further research is the concept of public aid for the healthcare industry in future pandemics, as the aid employed in Poland was a failure in terms of effectiveness and efficiency.

## 5. Conclusions

The shock of the COVID-19 pandemic has not only caused dramatic damage to people’s health and lives but also considerably influenced economies worldwide. Although significant work has already been conducted in respect of studying the impact of the pandemic on many industries, we consider the healthcare industry during the pandemic has not yet received enough attention from researchers. The healthcare industry is especially interesting from a scientific point of view as it was simultaneously affected by the pandemic in two opposite ways. As a result, some medical staff were significantly overloaded (e.g., infectious diseases, internal medicine, diagnostics), while other medical professionals, especially in small and medium-sized outpatient clinics, such as dentists or dermatologists, faced considerably reduced demand for their services. The objective of this paper was to study the efficiency of medical staff allocation and the performance of small and medium-sized outpatient clinics in Poland during the COVID-19 pandemic. We have identified that a number of small and medium-sized outpatient clinics, despite considerable decreases in the services provided and consequent deterioration of financial results, did not move into COVID-19 healthcare activities. We have identified that a considerable number of medical staff, despite possessing extensive medical knowledge, although not in infectious diseases, who were willing to provide healthcare services during the pandemic remained unused. Furthermore, we have identified that 4 of the 5 cases were passive or nearly passive against the pandemic and received donations, such as other non-healthcare businesses. Therefore, in view of future pandemics, we recommend considering development of a separate public aid system dedicated solely to the healthcare sector and aimed at enabling the shift of medical personnel from less occupied healthcare entities to more occupied ones.

## Figures and Tables

**Table 1 ijerph-19-14827-t001:** Case 1 financial results for 2019–2021, changes in financial measures within the 3-year period as absolute figures and as percentages. All amounts in EUR thousand.

	2019	2020	2021	Change	% Change
Dentistry	2316.3	2006.1	3518.2	1202	52%
Primary healthcare	1008.7	1149.0	1225.0	216	21%
Specialized health care	938.3	498.4	680.7	−258	−27%
Mobile dentistry	343.2	238.8	460.3	117	34%
COVID-19 testing		1861.0	12,215.5	12,216	
COVID-19 dentistry		638.5	331.1	331	
Vaccinations			1502.1	1502	
**Sales revenues**	**4606.5**	**6391.8**	**19,933.1**	**15,327**	**333%**
medical doctors’ salaries	−1963.0	−1800.1	−2964.7	−1002	51%
medical personnel salaries	−572.1	−1038.6	−2691.4	−2119	370%
medical services	−298.8	−666.0	−2065.1	−1766	591%
materials	−177.2	−356.2	−1236.3	−1059	598%
**Medical margin**	**1595.5**	**2530.9**	**10,975.6**	**9380**	**588%**
Remaining running costs	−1347.0	−1768.7	−4276.6	−2930	217%
**Profit before tax**	**248.4**	**762.2**	**6699.0**	**6451**	**2597%**
*as a % of sales*	*5.4%*	*11.9%*	*33.6%*		
*Inflation rate (year to year)*	*2.3%*	*3.4%*	*5.1%*		

**Table 2 ijerph-19-14827-t002:** Case 2 financial results for 2019–2021, changes in financial measures within the 3-year period as absolute figures and as percentages. All amounts in EUR thousand.

	2019	2020	2021	Change	% Change
Dentistry	259.9	247.6	241.5	−18	−7%
Primary healthcare	277.7	271.0	275.1	−3	−1%
Specialized health care	68.1	51.5	52.2	−16	−23%
Vaccinations			59.4	59	
**Sales revenues**	**605.8**	**570.1**	**568.8**	**−37**	**−6%**
medical doctors’ salaries	−255.7	−238.3	−261.1	−5	2%
medical personnel salaries	−87.4	−94.3	−118.9	−31	36%
medical services	−48.1	−38.6	−47.3	1	−2%
materials	−25.4	−20.5	−18.7	7	−26%
**Medical margin**	**189.2**	**178.4**	**122.9**	**−66**	**−35%**
Remaining running costs	−151.5	−156.0	−200.3	−49	32%
**Profit before tax**	**37.7**	**22.4**	**−77.4**	**−115**	**−305%**
*as a % of sales*	6.2%	3.9%	−13.6%		
*Inflation rate (year to year)*	*2.3%*	*3.4%*	*5.1%*		

**Table 3 ijerph-19-14827-t003:** Case 3 financial results for 2019–2020, changes in financial measures within the 2-year period as absolute figures and as percentages. All amounts in EUR thousand.

	2019	2020	Change	% Change
Dentistry	331.1	251.4	−79.8	−24%
**Sales revenues**	**331.1**	**251.4**	**−79.8**	**−24%**
medical doctors’ salaries	−152.4	−110.5	41.9	−27%
medical personnel salaries	−39.6	−39.4	0.2	−1%
medical services	−25.2	−20.0	5.2	−21%
materials	−20.3	−20.7	−0.5	2%
**Medical margin**	**93.6**	**60.7**	**−32.9**	**−35%**
Remaining running costs	−53.1	−32.6	20.5	−39%
**Profit before tax**	**40.5**	**28.1**	**−12.4**	**−31%**
*as a % of sales*	*12.2%*	*11.2%*		
*Inflation rate (year to year)*	*2.3%*	*3.4%*		

**Table 4 ijerph-19-14827-t004:** Case 4 financial results for 2019–2021, changes in financial measures within the 3-year period as absolute figures and as percentages. All amounts in EUR thousand.

	2019	2020	2021	Change	% Change
Dentistry	802.7	610.0	936.9	134.2	17%
**Sales revenues**	**802.7**	**610.0**	**936.9**	**134.2**	**17%**
medical doctors’ salaries	−292.9	−214.8	−344.4	−51.5	18%
medical personnel salaries	−65.5	−47.5	−83.5	−18.0	27%
medical services	−44.6	−38.5	−62.8	−18.2	41%
materials	−121.3	−95.0	−138.4	−17.1	14%
**Medical margin**	**278.5**	**214.3**	**307.7**	**29.3**	**11%**
Remaining running costs	−205.7	−210.0	−238.8	−33.1	16%
**Profit before tax**	**72.7**	**4.3**	**68.9**	**−3.8**	**−5%**
*as a % of sales*	*9.1%*	*0.7%*	*7.4%*		
*Inflation rate (year to year)*	*2.3%*	*3.4%*	*5.1%*		

**Table 5 ijerph-19-14827-t005:** Case 5 financial results for 2019–2021, changes in financial measures within the 3-year period as absolute figures and as percentages. All amounts in EUR thousand.

	2019	2020	2021	Change	% Change
Dentistry	294.0	242.9	391.7	97.6	33%
**Sales revenues**	294.0	242.9	391.7	97.6	33%
medical doctors’ salaries	−105.3	−96.3	−140.0	−34.7	33%
medical personnel salaries	−41.7	−43.3	−57.4	−15.7	38%
medical services	−19.1	−18.6	−28.1	−9.0	47%
materials	−18.1	−22.0	−29.5	−11.4	63%
**Medical margin**	109.8	62.7	136.6	26.9	24%
Remaining running costs	−65.3	−63.5	−96.6	−31.3	48%
**Profit before tax**	44.5	−0.8	40.1	−4.4	−10%
*as a % of sales*	*15.1%*	*−0.3%*	*10.2%*		
*Inflation rate (year to year)*	*2.3%*	*3.4%*	*5.1%*

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
