# Peer review of "Medical Staff Shortages and the Performance of Outpatient Clinics in Poland during the COVID-19 Pandemic"

_ijerph, 2022, doi:10.3390/ijerph192214827_

Round 1
Reviewer 1 Report
please find enclosed the manuscript with my comments and remarks

Author Response
Dear Reviewer,
Thank you very much for time spent on our manuscript and your valuable indications. We have implemented each of your suggestions.
Additionally, we would like to note, with respect to the followings:
- The language has been improved by native speaker through the whole text
- Maintaining infection prevention and control measures for COVID-19 in health care facilities. Policy brief. 7 June 2022 – as indicated the paper has been cited (see lines 40-49)
- Mobile dentistry (your indication for Table 1) has been explained better, see lines: 135-137
- We provided more statements and quotes in results section, supporting our conclusion at lines 487-490 (before adjustments 471-473).
- We also added Appendix A, providing key information, in a similar way for each of studied cases
- Finally, at conclusions section we added more specific recommendations (see lines 607-610)
Thank you again and kind regards,
The Authors
Reviewer 2 Report
Dear Authors, I read with interest your manuscript. The proposed method is interesting, as well as the analysis of the COVID impact on small healthcare facilities in economic terms. However, please consider my comments:
- Lines 55-61 should be rewritten. In fact, the healthcare industry has not been “selected”, but is the investigated entity.
- English should be improved.
- Lines 97-101 should be removed.
- Lines 120-125: the selection criteria should be discussed and motivated
- Description of “Cases” (clinics) should be made similar between each other
- Line 186: the vaccination status of interviewer/interviewed and the mask use do not add any useful information. Please remove.
- Financial data: personally, I agree to avoid multiple exchange rate to avoid confusion. However, I suggest to add some information about the inflation during the studied period. If it was significant (but I do not think so), results should be discussed taking it into account.
- Line 218: I suggest to use “assessed” instead of “triangulated”.
- Tables of financial results: I suggest to express profit before tax also as percentage on total revenues
- Structure of the companies should be explained better. I suggest to add a table summarizing the most important information. Furthermore, the five financial results tables could be embedded in a single one, or published as Appendix.
- Could be interesting to add quotes from employees in all the five cases (e.g., case 5 do not have any quote)
- Discussion: At line 409, I suggest to better introduce the sentence.
- Lines 425-432: these numerical results should be stated in the “results” section, not in the discussion. The discussion should contain only the interpretation of the previous sections.
- The Authors correctly stated the limitations of the manuscript. However, I suggest to highlight the qualitative approach used in the research.
Author Response
Dear Reviewer,
Thank you very much for time spent on our manuscript and your valuable indications. We have implemented each of your suggestions.
Our detailed responses are as follows:
- Lines 55-61 should be rewritten. In fact, the healthcare industry has not been “selected”, but is the investigated entity - done, see line 59
- English should be improved. – done, through the whole manuscript
- Lines 97-101 should be removed - done
- Lines 120-125: the selection criteria should be discussed and motivated - done, see lines 118-124
- Description of “Cases” (clinics) should be made similar between each other – several adjustments have been introduced and additionally Appendix A with key information for each of studied cases
- Line 186: the vaccination status of interviewer/interviewed and the mask use do not add any useful information. Please remove. - done
- Financial data: personally, I agree to avoid multiple exchange rate to avoid confusion. However, I suggest to add some information about the inflation during the studied period. If it was significant (but I do not think so), results should be discussed taking it into account. - done
- Line 218: I suggest to use “assessed” instead of “triangulated”. - done
- Tables of financial results: I suggest to express profit before tax also as percentage on total revenues - done
- Structure of the companies should be explained better. I suggest to add a table summarizing the most important information. Furthermore, the five financial results tables could be embedded in a single one, or published as Appendix. Done, please see Appendix A with key information for each of studied cases
- Could be interesting to add quotes from employees in all the five cases (e.g., case 5 do not have any quote) - done
- Discussion: At line 409, I suggest to better introduce the sentence. - done, see line 428
- Lines 425-432: these numerical results should be stated in the “results” section, not in the discussion. The discussion should contain only the interpretation of the previous sections. - done
- The Authors correctly stated the limitations of the manuscript. However, I suggest to highlight the qualitative approach used in the research. - done, see lines 572-585
Thank you again and kind regards,
The Authors
Round 2
Reviewer 2 Report
Dear Authors,
thank you for addressing my suggestions, much appreciated. I think that now the manuscript deserves the publication. However, one last minor point. In lines 586-587 and 607-609 you are very critic toward the Polish government. I suggest to "mitigate" these sentences, avoiding a direct comment to the government that can not be deducted from the manuscript.
Author Response
Dear Reviewer,
Thank you very much for revision of our paper and your following indications regarding political correctness of presented conclusions. As indicated the conclusions have been accordingly presented (see lines 605-608, in track changes mode). In order to remain consistent, we deleted one sentence from the abstract (see line 16).
Kind regards,
The authors,